# Hemispherotomy in Infants with Hemimegalencephaly: Long-Term Seizure and Developmental Outcome in Early Treated Patients

**DOI:** 10.3390/brainsci13010073

**Published:** 2022-12-30

**Authors:** Chiara Pepi, Alessandro De Benedictis, Maria Camilla Rossi-Espagnet, Simona Cappelletti, Martina Da Rold, Giovanni Falcicchio, Federico Vigevano, Carlo Efisio Marras, Nicola Specchio, Luca De Palma

**Affiliations:** 1Rare and Complex Epilepsies Unit, Department of Neuroscience, Bambino Gesù Children’s Hospital, IRCCS, Full Member of European Reference Network EpiCARE, 00165 Rome, Italy; 2Neurosurgery Unit, Bambino Gesù Children’s Hospital, IRCCS, 00165 Rome, Italy; 3Neuroradiology Unit, Imaging Department, Bambino Gesù Children’s Hospital, IRCCS, 00165 Rome, Italy; 4Unit of Clinical Psychology, Department of Neuroscience, Bambino Gesù Children’s Hospital, IRCCS, 00165 Rome, Italy; 5Scientific Institute, IRCCS “E. Medea”, Association “La Nostra Famiglia”, 31015 Conegliano, Italy; 6Department of Basic Medical Sciences, Neurosciences and Sense Organs—University of Bari Aldo Moro, 70121 Bari, Italy

**Keywords:** hemimegalencephaly, epilepsy surgery, hemispherotomy, developmental outcome

## Abstract

Hemimegalencephaly (HME) is a rare brain congenital malformation, consisting in altered neuronal migration and proliferation within one hemisphere, which is responsible for early onset drug-resistant epilepsy. Hemispherotomy is an effective treatment option for patients with HME and drug-resistant epilepsy. Surgical outcome may be variable among different surgical series, and the long-term neuropsychological trajectory has been rarely defined using a standardized neurocognitive test. We report the epileptological and neuropsychological long-term outcomes of four consecutive HME patients, operated on before the age of three years. All patients were seizure-free and drug-free, and the minimum follow-up duration was of five years. Despite the excellent post-surgical seizure outcome, the long-term developmental outcome is quite variable between patients, ranging from mild to severe intellectual disabilities. Patients showed improvement mainly in communication skills, while visuo-perceptive and coordination abilities were more impaired. Epileptological outcome seems to be improved in early treated patients; however, neuropsychological outcome in HME patients may be highly variable despite early surgery.

## 1. Introduction

Hemimegalencephaly (HME) is a rare malformation of cortical development characterized by an abnormal neuronal proliferation, with enlargement of one cerebral hemisphere [1].

The abnormal hemisphere usually shows abnormal gyrification, thick cortex, ventricular asymmetry, abnormal gray–white matter differentiation, neuronal heterotopia, basal ganglia, and internal capsule abnormalities [2]. HME may occur as an isolated disease or as a part of different neurocutaneous syndromes [3].

Epileptic seizures occur in up to 93% of patients with HME, and onset of seizures is usually before the age of six months. Soon after, the onset seizures become drug-resistant with multiple daily seizures. Epilepsy in patients with HME is frequently associated with developmental delay and focal neurological deficit, as contralateral hemiparesis and hemianopsia [4]. 

HME does not show familial inheritance or sex preference [3]. Somatic genetic variants in the PI3K-Akt-mTOR signaling pathway, such as AKT3, PIK3CA, MTOR, and DEPDC5, have been reported [5,6,7]. 

A surgical approach should be considered very soon in patients with HME and drug-resistant epilepsy. Hemispherotomy is the best surgical technique, with lateral or parasagittal approaches [8]. The reported seizure freedom rates in hemispherotomy cohorts range from 50% to 90% [9,10,11,12,13], with some discrepancies seen in different studies with different follow-up duration.

Factors affecting seizure outcome are still not completely understood. HME, as developmental etiology, is a negative predictor for seizure control after surgery [14,15].

Subtle bilateral alterations in Magnetic Resonance Imaging (MRI) and contralateral FDG Positron Emission Tomography (FDG-PET) hypometabolism [16,17,18] may predict seizure recurrence after surgery. Surgical technique (lateral vs. parasagittal hemispherotomy) seems not to influence the post-operative outcome [11,12,19].

Seizure freedom after hemispherotomy may positively influence the developmental outcome [13,20,21].

Only a few studies [4,22,23,24,25] have investigated developmental outcome in HME patients, with some limitations due to small sample size and the lack of standardized testing.

Results have shown that 70–92% of HME operated patients have moderate to severe intellectual disability in the long-term follow-up [4,24,26]. Post-operative improvement in developmental level is occasionally reported for some children [4,26].

The aim of this report is to describe the long-term (up to five years) longitudinal neuropsychological profile, through standardized neuro-cognitive testing, and the epileptological outcomes of four consecutive patients with drug-resistant epilepsy associated with HME who underwent early hemispherotomy.

## 2. Methods

### 2.1. Patients’ Selection

We performed a retrospective review of all pediatric patients (<18 years) who underwent hemispherotomy for drug-resistant epilepsy from January 2009 to December 2016, in Bambino Gesù Children’s Hospital, Rome, Italy. This study was approved by the local ethics committee. The local ethics committee waived the written informed consent for collection of these data from retrospective review of records.

We found 19 patients, and we collected their clinical, neuroradiological, and neuropathological data. Among them, we filtered patients with histopathologic diagnosis of HME and minimum follow-up duration of five years. Four patients were selected. These patients were already reported as aggregate data in a previous multicenter publication [27].

### 2.2. Presurgical Assessment

All patients underwent routine pre-surgical evaluation, including full history and neurological examination, visual analysis of long-term video-EEG monitoring with multiple typical seizures recorded and 3 Tesla brain MRI. Seizures were classified according to the ILAE Position Paper for Classification and Terminology [28]. Suitability for surgery was discussed during multi-disciplinary epilepsy surgery meetings. The decision to offer surgery was based on predicted seizure outcome from pre-surgical data and surgical risks.

All patients underwent neuropsychological assessment before surgery and during follow-up.

We analyzed pre- and post-surgical developmental trajectories, considering Wechsler Scale (WISC-IV or WPPSI-III) and Griffiths Mental Development Scale (GMDS), and we obtained three different quotients [29,30,31]:

A General Quotient, including the mean of Total Intelligence Quotient in Wechsler Scale and Developmental Quotient in GMDS; 

A Verbal Quotient, including the mean of Verbal Intelligence Quotient in Wechsler Scale and C scale (Eye and Hand Coordination) in GMDS;

A Performance Quotient, including the mean of Performance Intelligence Quotient in WPPSI-III, Perceptual Reasoning Index in WISC-IV, and mean of D-E-F domains in GMDS. 

Two patients (#3 and #4), operated on at 3 and 2 months, did not perform a pre-surgical developmental level because at the time of surgery they were not testable due to lack of motor or language acquisitions.

## 3. Results

Four patients were included in this study. Age at seizure onset was between 1 and 5 months of age (median 2 months). One patient (Case #1) presented with asymmetric spasms, two patients (Case #2 and #3) with focal tonic seizures, and one patient (Case #4) with focal clonic seizures. Age at surgery was between 2 months and 2 years and 9 months (median 9.5 months). All patients received hemispherotomy with the vertical parasagittal approach. Neurological follow-up duration was between 5 and 11 years (median 6.5 years).

All patients were seizure-free at last follow-up, and they have withdrawn anti-seizure medications (ASM) (Engel 1a) [32]. Three post-surgical complications occurred: hydrocephalus in two patients and transverse sinus thrombosis in one, without radiological and clinical long-term consequences. 

Main clinical findings are summarized in Table 1.

## 4. Case Reports

### 4.1. Patient #1

This is a 13-year-old female patient. She was born at term after an uneventful pregnancy, with normal delivery. Asymmetric tonic spasms started during the first week of life. Spasms were isolated or in cluster, involving mainly the left arm, with left eye deviation. Seizures were immediately multiple per day. The introduction of vigabatrin was initially efficacious. At the age of 3 months, left hemiparesis became evident. Asymmetric tonic spasms recurred at the age of 7 months.

Brain MRI revealed enlargement of the right posterior temporo-parietal and occipital region, with a gray–white matter blurring extending beyond the motor cortex, suggestive of HME (Figure 1a).

Long-term video-EEG monitoring showed an interictal activity of continuous pseudo-periodic spikes/polyspikes and slow waves, intermingled with low voltage fast activity over the right hemisphere, mainly in the posterior perisylvian region (Figure 2a). Multiple focal asymmetric spasms were recorded. EEG was characterized by a diffuse slow wave preceded by a high amplitude spike over the right temporal and parietal regions. 

At the age of 1 year and 4 months, she underwent parasagittal right hemispherotomy. 

Three years after surgery, she developed hydrocephalus that required ventriculoperitoneal shunt. After a follow-up duration of 11 years, she is seizure-free, ASMs were completely withdrawn at the age of 5 years, and EEG showed the persistence of a continuous pseudo-periodic paroxysmal pattern over the right hemisphere, without diffusion to the contralateral hemisphere. She has left hemiparesis (Medical Research Council grading system for strength: arm and leg 4/5), contralateral hemianopia, distractibility, and shortness in attention span. 

The Griffiths Mental Developmental Scales (GMDS), performed before surgery at 16 months, showed mild deficit—Developmental Quotient (DQ) 62—with major difficulties in Performance tasks (Appendix A).

The neuropsychological evaluation, with WISC IV scale, performed at 10.5 years of age (nine years after surgery), revealed a borderline cognitive functioning (full scale IQ 73, VIQ 92, PIQ 74) and difficulties in phonological memory, visual-perceptual analysis, and in constructive practices (Appendix A)

Figure 3a shows the comparison of General, Verbal, and Performance Domains between pre- (GMDS at 16 months of age) and post-surgical developmental level (WISC-IV at 10.5 years). She improved both in linguistic and performance abilities nine years after surgery, with a normal Verbal Quotient for her chronological age (92), while General and Performance Quotients were at borderline level (73 and 74, respectively).

### 4.2. Patient #2

This is an 11-year-old female patient born at term, with normal delivery. Focal hemi-tonic seizures, with main involvement of the left leg started during the second week of life. Soon after epilepsy onset, tonic asymmetric spasms were also evident. Multiple ASMs did not control seizure. Left hemiparesis was evident. Brain MRI (Figure 1b) showed a right, mainly posterior, HME. Parents initially refused the surgical procedure. EEG showed a continuous slow and paroxysmal activity over the right posterior perisylvian region.

Hemispherotomy was performed at the age of two years and nine months. One month after surgery, she presented post-surgical hydrocephalus treated with ventriculoperitoneal shunt. 

After seven years of follow-up (10 y/old), she is seizure-free and drug-free (last ASM withdrawn, at the age of eight). She presents with left hemiparesis (MRC arm and leg 3/5). The post-operative EEG showed the persistence of epileptiform activity only over the right hemisphere.

GMDS, pre-surgically performed at 30 months of age, revealed a moderate deficit (DQ 53) with homogenous subscales (Appendix A). Two years after surgery (5 y/old), she performed a new GMDS (Appendix A), showing a stable general developmental level with improvement in communication skills, both verbal and non-verbal. Motor performances were clearly worsened by the post-surgical hemiparesis. 

Figure 3b shows pre- and post-surgical developmental quotients, configuring a global cognitive impairment, quite homogenous among different domains.

Three years after surgery (six years of age), a non-verbal cognitive evaluation was performed (Leiter-R scale, due to linguistic limitations). She obtained a brief IQ of 62 and a Fluid Reasoning Index of 65.

The adaptive functioning was tested after a follow-up duration of five years (eight years of chronological age), with Vineland Survey Form, showing a moderate adaptive difficulty (Adaptive Quotient of 49), with daily living problems. 

### 4.3. Patient #3 

This is an 8-year-old male child, with epileptic seizures having started in the first week of life. Seizures were asymmetric focal tonic spasms involving mainly the right arm. Brain MRI revealed a large HME (Figure 1c) over the left temporo-parieto-central cortex, and EEG showed a pseudo-periodic paroxysmal pattern involving the left hemisphere, mainly over temporal and parietal regions (Figure 2b). Seizures were focal motor with a tonic posturing of the right arm (Figure 2c). Multiple ASMs were tried without effect.

Surgery was performed at the age of 3 months. No clear hemiparesis was evident before the procedure. No complication occurred. EEG during follow-up showed paroxysmal activity confined to the left hemisphere with no seizure recurrence. The neurological examination evidenced right hemiparesis with recovery of antigravity movement during the follow-up (MRC 3/5). The neuropsychological evaluation was not performed before surgery, considering that at three months he was not able to control his head nor look at the examiner. Five years after surgery, he is seizure-free tapering completely all drugs. Seven years after surgery (7y/old), his neuropsychological evaluation with the Wechsler Scale (WISC-IV) showed a TIQ of 49, VIQ of 56, and PIQ of 85 (Figure 3c), configuring a moderate intellectual disability.

### 4.4. Patient #4

This is a 5-year-old boy, born at term after an uneventful pregnancy. His neurologic examination showed poor visual fixation, diffuse hypotonus, and left hemiparesis, with homologous hemianopsia. Epilepsy started a few days after birth, with multiple left clonic seizures per day. MRI evidenced a right HME (Figure 1d). Interictal EEG was characterized by continuous slow delta waves over the right occipital area, sometimes diffused over the homologous contralateral area. Frequent subclinical rhythmic discharges were recorded over the right occipital regions, sometimes with left diffusion (Figure 2d). During sleep, the left hemisphere showed a physiologic sleep pattern while the right one preserved a pathological activity with unilateral discharges.

The neuropsychological evaluation was not performed before surgery, considering that at two months he was not able to control his head nor look at the examiner. 

Hemispherotomy was performed at two months of age. He presented after surgery a transverse sinus thrombosis, successfully treated with anticoagulant therapy. One month after surgery, he performed a GMDS at 3.5 months, followed by three more developmental scales (at 15, 27, and 48 months of age). Detailed results are showed in Appendix A.

After surgery, he presented a progressive improvement in global active movements, especially in the left side and in the attention towards the left visual space. 

He tapered off ASMs one year after surgery, and after a follow-up of five years he is seizure-free.

At five years of age, he walked with support and started pronouncing some disyllables.

Figure 3d shows mild modification of Verbal and Performance Quotients after surgery, with mean values lower than 40 at last follow-up. 

The patient showed a de novo pathogenetic constitutional nonsense genetic variant (NM_000314.7 c.77delC) in the PTEN gene.

## 5. Discussion

The aim of our study is to describe the long-term epileptological and developmental outcome, evaluated through direct and standardized cognitive testing, in patients with HME who underwent early surgical treatment (before three years of age).

Previously reported seizure freedom rates in hemispherotomy cohorts range from 50% to 90% [9,10,11,12,13], including studies with mixed etiologies and different follow-up duration. Developmental etiology, as HME, is associated with a worse post-surgical epileptological outcome [14,33]. Results show that 60–70% of children with HME are seizure-free after surgery [10,34]. In our cohort, all patients are seizure-free after a mean follow-up of 6.5 years, and they have withdrawn ASMs.

The excellent post-surgical outcomes reinforce the idea that surgery should be considered soon in patients affected by wide hemispheric pathology [4,10].

The cognitive evolution of these patients is heterogeneous. Previously reported studies suffered from some limitations in sample size and/or quality reporting of cognitive outcomes. However, in the long-term evaluation after surgery, 70–92% of patients have severe or moderate intellectual disability [4,24,26].

Factors possibly related to better developmental outcome include seizure freedom after surgery [24], lower duration of epilepsy before surgery [20,24], and lack of contralateral hemispheric malformations on brain MRI [35]. More recently, a left localization of HME was found to have poorer cognitive and language outcomes [25,36].

There is a correlation between poor pre-operative developmental level and post-surgical cognitive delay [20,34,37,38]. 

In our case series, we evidenced that despite seizure freedom, cognitive trajectories may be quite different. Our first three patients showed an improvement after surgery, up to a mild/borderline intellectual disability, while the last one presented less significant improvement after surgery. This patient carried a constitutional pathogenetic variant in the PTEN gene which may be responsible for the less favorable neurocognitive profile after surgery. In this last case, the delay in cognitive and adaptive capabilities became more and more clear during the follow-up. 

A better pre-surgical cognitive level is clearly associated with good cognitive post-surgical level in our first two patients, consistently with what is already reported [20,22,34]. Probably, not all HME patients have the same cognitive “reservoir” in the health hemisphere, and genetic constitutional variants may contribute to more severe neurocognitive profile. 

We cannot assess the influence of the timing of surgery, as even patients operated on early (patient #3 and #4) showed a heterogeneous cognitive outcome. There is no agreement about the role of epilepsy duration in cognitive functions in patients with HME [25].

We also noted that after surgery (patients #1 and #2) there might be an improvement in communication and performance skills, although visuo-perceptive and coordination abilities may be impaired by subsequent hemiparesis.

Our study has major limitations due to retrospective recruitment and the low sample size. Despite these limitations, we may consider our experience as a proof of concept that early surgery in HME is feasible and highly successful. Patients with HME may have delayed developmental milestones after surgery, sometimes becoming more clearcut over years. Profound cognitive impairment may be related to constitutional genetic variants. Patients constantly require significant language and cognitive long-term support, especially on visuo-perceptive and coordination abilities. 

Parasagittal hemispherotomy is an effective surgical procedure for hemispheric pathologies. We can reach good seizure control in infants with HME. Neuropsychological trajectories in HME patients may be characterized by moderate to severe intellectual disability, despite good seizure control. Further studies are needed to better delineate a genetic–neuropsychological correlation.

## Figures and Tables

**Figure 1 brainsci-13-00073-f001:**
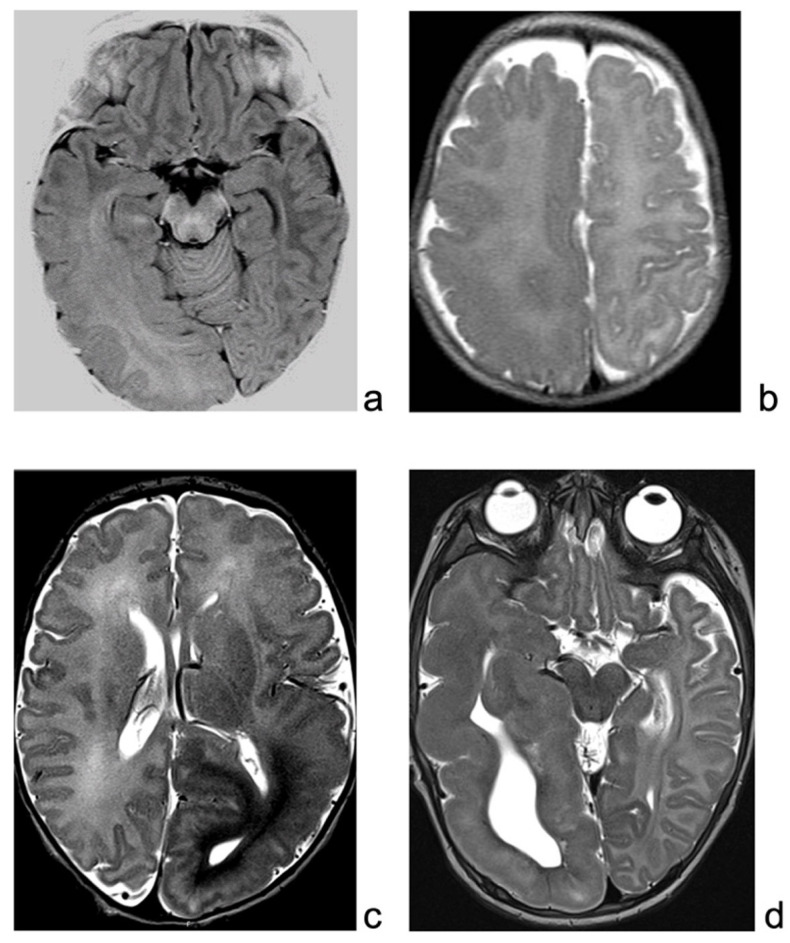
(**a**): MRI of patient 1, showing enlarged right hemisphere, with gray–white matter blurring mainly in the temporo-occipital regions. (**b**): T2-weighted sequence of patient 2 MRI, showing right enlargement of temporo-occipital regions, with blurring, extending also over the suprasylvian regions. (**c**): MRI of patient 3, showing enlarged and malformed cortex over the left hemisphere, mainly posteriorly. (**d**): MRI of patient 4, T2-weighted sequence, showing enlargement and malformation of left hemisphere, mainly temporal region.

**Figure 2 brainsci-13-00073-f002:**
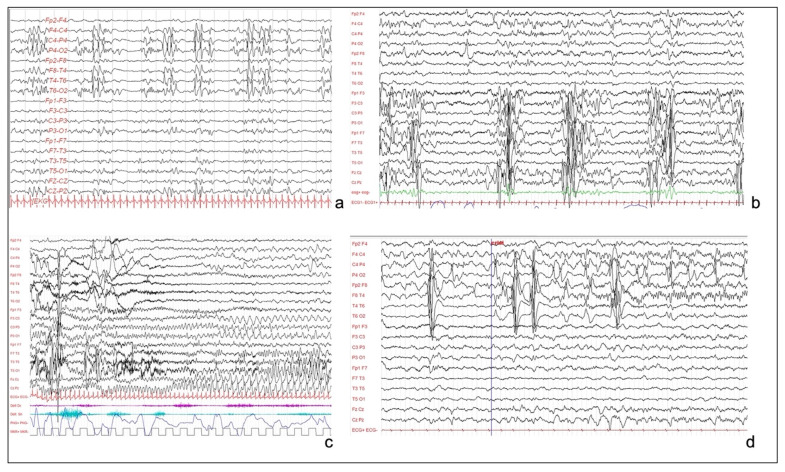
(**a**): Interictal EEG of patient 1, showing sub-continuous poly-sharp waves and slow waves with some flattening over the right posterior perisylvian region. (**b**): Interictal EEG of patient 3, showing pseudo-periodic paroxystic pattern involving the left hemisphere, characterized by repetitive polyspikes. (**c**): Patient 3. Focal seizure, characterized by rhythmic theta activity involving the left hemisphere and the vertex, without contralateral diffusion. (**d**): EEG of patient 4, showing continuous and pseudo-periodic spike–wave complexes over the right centro-temporal derivations, with focal seizure starting from central and temporal regions.

**Figure 3 brainsci-13-00073-f003:**
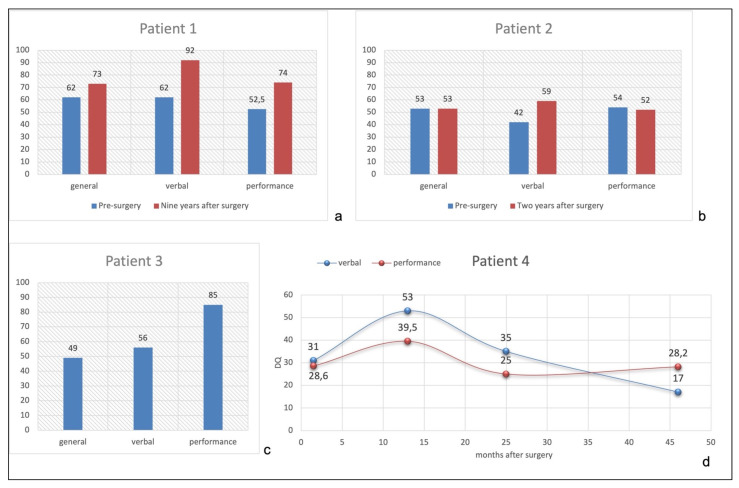
(**a**): Patient 1 pre- and post-surgical quotients. (**b**): Patient 2 pre- and post-surgical quotients. (**c**): Patient 3 post-surgical quotients. (**d**): Patient 4 post-surgical verbal and performance quotients performed at 3.5, 15, 27, and 48 months after surgery.

**Table 1 brainsci-13-00073-t001:** Summary of clinical and epileptological patients’ data.

Pts	Age at Seizure Onset (Years)	Age at Surgery (Years)	Anti-Seizure Medications	MRI Findings	Seizure Semiology	EEG Findings	Outcome	Neurologic Follow-Up (Years)	Post-Surgical Complications
1	0.3	1.5	GVG	Right HME	Asymmetric spasms (left)	Continuous spikes/polyspikes and slow waves, with low voltage fast activity over the right hemisphere.	Engel Ia	11	Hydrocephalus
2	0.5	2.9	PB, ACTH, VPA, GVG, CLB, LEV	Right HME	Left tonic seizures and asymmetric spasms	Continuous periodic paroxysmal activity over the right posterior perisylvian region.	Engel Ia	7	Hydrocephalus
3	0.1	0.3	PB, GVG, ACTH, CBZ	Left HME	Right asymmetric spasms and tonic seizures	Pseudo-periodic paroxysmal pattern over the left hemisphere, mainly over temporal and parietal regions.	Engel Ia	5	No
4	0.1	0.2	VGB, PB, LEV	Right HME	Left clonic seizures	Continuous slow delta waves over the right occipital area, sometimes diffused over the homologous contralateral area.	Engel Ia	5	Left transverse venous sinus thrombosis.

ASM: anti-seizure medications; CBZ: carbamazepine, CLB: clobazam, LEV: levetiracetam; HME: hemimegalencephaly; PB: phenobarbital; VGB: vigabatrin; VPA: valproic acid.

## Data Availability

Not applicable.

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
