# Peer review of "Hemispherotomy in Infants with Hemimegalencephaly: Long-Term Seizure and Developmental Outcome in Early Treated Patients"

_brainsci, 2022, doi:10.3390/brainsci13010073_

Round 1

Reviewer 1 Report

Comments and Suggestions for Authors

This article is reporting retrospective analysis of four young patients affected by hemimegalencephaly with epileptic seizures, who underwent brain surgery in early developmental ages. The analysis suggest that the brain surgery is effective in treating seizures, and rescue some cognitive ability, however affecting other with individual variability.

Overall this paper is of clinical interest, and well written directed to a niche audience.

In the methods the authors can be more specific about the fact  that the study is  retrospective. There is no mention of Ethical approval, and when there are patients tissue/data reported in a paper the Ethics is a must. Strong advice to add it.

Author Response

REVIEWER 1

Q: This article is reporting retrospective analysis of four young patients affected by hemimegalencephaly with epileptic seizures, who underwent brain surgery in early developmental ages. The analysis suggest that the brain surgery is effective in treating seizures, and rescue some cognitive ability, however affecting other with individual variability.

Overall this paper is of clinical interest, and well written directed to a niche audience.

In the methods the authors can be more specific about the fact that the study is  retrospective. There is no mention of Ethical approval, and when there are patients tissue/data reported in a paper the Ethics is a must. Strong advice to add it.

A: We thank this reviewer for these comments. According to your suggestions, we clearly specified in the Methods section the retrospective nature of this study: “We performed a retrospective review of all pediatric patients (<18 years) who underwent vertical parasagittal hemispherotomy for drug-resistant epilepsy from January 2009 to December 2016, in Bambino Gesù Children Hospital, Rome, Italy”.

We further already requested Ethic Committee approval for anonymous publications of patients’ data, and we modified the text as follows: “This study was approved by the local ethics committee. The local ethics committee waived the written informed consent for collection of these data from retrospective review of records”.

Reviewer 2 Report

Comments and Suggestions for Authors

The authors present an important observation in post-surgical cases of a rare condition of hemimegalencephaly. The paper is clinically relevant, well written and the authors acknowledge the limitations of the study.

Author Response

REVIEWER 2

Q: The authors present an important observation in post-surgical cases of a rare condition of hemimegalencephaly. The paper is clinically relevant, well written and the authors acknowledge the limitations of the study.

A: We kindly thank this reviewer for your appreciation.

Reviewer 3 Report

Comments and Suggestions for Authors

The manuscript is nicely written, and the results are well-displayed.

Major concern is that I cannot find any new data in the manuscript. It is mere report ion 4 cases, which is (even though rare disease) a very low number and might explain the seizure free rate of 100%.

Long term neuro cognitive followup in this patient cohort has just been reported (45 patients) in 2021. 

In my eyes this additional report does not add substantially to the existing literature.

Author Response

REVIEWER 3

Q: The manuscript is nicely written, and the results are well-displayed.

Major concern is that I cannot find any new data in the manuscript. It is mere report ion 4 cases, which is (even though rare disease) a very low number and might explain the seizure free rate of 100%.

Long term neuro cognitive follow-up in this patient cohort has just been reported (45 patients) in 2021.

In my eyes this additional report does not add substantially to the existing literature.

A: We thank the reviewer for explaining his point. We fully agree with the reviewer about the strong limitation of our study due to the low number of patients. Surely, HME is a rare pathology and we decided to evaluate only patients who underwent early surgery and were longitudinally followed.

The previous paper mentioned by the reviewer (Puka et al., Epilepsia 2021), include 45 HME patients evaluated though parent-reports of cognitive and language skills. We cited this work in our study and we believe that our study might add a direct quantitative cognitive test, through Griffiths or Wechsler Scales, that were never systematically performed before in HME patients, to our knowledge.

We subsequently underlined in the text this point, adding that patients were “evaluated though direct and standardized cognitive test

Reviewer 4 Report

Comments and Suggestions for Authors

The manuscript is really a case series of four patients. But anyway important due to the detailed description of the long time cognitive outcomes. In spite of being seizure free there may still be severe cognitive disabilities. I think this is an important description for pediatric neurologists and pediatric epilepsy surgeons. There are some typing errors though:

Page 2 row 51 the sentence Eipilepsy in patiens with HME is frequently be associate. Needs to be corrected.

The legend to Figure 1: In D I guess it should be patient number 4 not number 3.

The same error in the legend for Figure 2. I guess in 2B it should be patient number 2 not number 3.

In page 8, row 311"is reported already reported" needs to be corrected.

Then the legend for the supplementary Table should be corrected. I guess table 1 is for patient number 1, not stated, in table 2 it says table number 1, is that really so. int table 3 it says patient 2, is that correct. In table 4 it is probably corret stated patient number 4.

Author Response

REVIEWER 4

Q: The manuscript is really a case series of four patients. But anyway important due to the detailed description of the long time cognitive outcomes. In spite of being seizure free there may still be severe cognitive disabilities. I think this is an important description for pediatric neurologists and pediatric epilepsy surgeons. There are some typing errors though:

A: We kindly thank the reviewer for highlighting some typing errors. We modified the text accordingly. Please see below:

  • Page 2 row 51 the sentence Epilepsy in patients with HME is frequently be associate. Needs to be corrected.
  • This was corrected as suggested.
  • The legend to Figure 1: In D I guess it should be patient number 4 not number 3.
  • This was corrected as suggested.
  • The same error in the legend for Figure 2. I guess in 2B it should be patient number 2 not number 3.
  • We thank this reviewer for this comment. In figure 2B we conform that is related to patient 3. We decided to include two EEG snapshots of this patient because we found his EEG more explicative.
  • In page 8, row 311"is reported already reported" needs to be corrected.
  • This was corrected as suggested.
  • Then the legend for the supplementary Table should be corrected. I guess table 1 is for patient number 1, not stated
  • This was corrected as suggested.
  • In table 2 it says patient number 1 is that really so.
  • We confirm that it is correct, supplementary Table 2 is also related to patient#1
  • In table 3 it says patient 2, is that correct.
  • We confirm that it is correct, supplementary Table 3 is related to patient#2
  • In table 4 it is probably correct stated patient number 4.
  • This is also correct

Round 2

Reviewer 3 Report

Comments and Suggestions for Authors

I ackowledge the authors' rebutal to my previous concern.

Nevertheless, given the very small field of interested readers and the extremely low number of indivduals, I would rathe rrecomment to publish it as case report/series. I am rather worried about the scientific significance given the outcome rates of 100%.

Author Response

We thank this reviewer for this comment. We acknowledged in the manuscript that this is a small series of patients, however being a rare epilepsy we feel this may considered representative of a wider number of patients. The outcome rate of 100% is also explained by the relatively low number of cases. Moreover a detailed and structured neurocognitve evaluation is not clearly reported and this may increase the awareness around this spesici topic.